# Vapor detection and discrimination with a panel of odorant receptors

Hitoshi Kida[1,2], Yosuke Fukutani [1,3], Joel D. Mainland [1,4,5], Claire A. de March[1], Aashutosh Vihani[1,6], Yun Rose Li[1,9], Qiuyi Chi[1], Akemi Toyama[1], Linda Liu[1], Masaharu Kameda [2,7], Masafumi Yohda [3,7] & Hiroaki Matsunami [1,6,7,8]

Olfactory systems have evolved the extraordinary capability to detect and discriminate volatile odorous molecules (odorants) in the environment. Fundamentally, this process relies on the interaction of odorants and their cognate olfactory receptors (ORs) encoded in the genome. Here, we conducted a cell-based screen using over 800 mouse ORs against seven odorants, resulting in the identification of a set of high-affinity and/or broadly-tuned ORs. We then test whether heterologously expressed ORs respond to odors presented in vapor phase by individually expressing 31 ORs to measure cAMP responses against vapor phase odor stimulation. Comparison of response profiles demonstrates this platform is capable of discriminating between structural analogs. Lastly, co-expression of carboxyl esterase Ces1d expressed in olfactory mucosa resulted in marked changes in activation of specific odorant-OR combinations. Altogether, these results establish a cell-based volatile odor detection and discrimination platform and form the basis for an OR-based volatile odor sensor.

[1] Department of Molecular Genetics and Microbiology, Duke University Medical Center, Durham, NC 27710, USA. [2] Department of Mechanical Systems Engineering, Tokyo University of Agriculture and Technology, Koganei, Tokyo 184-8588, Japan. [3] Department of Biotechnology and Life Science, Tokyo University of Agriculture and Technology, Koganei, Tokyo 184-8588, Japan. [4] Monell Chemical Senses Center, Philadelphia, PA 19104, USA. [5] Department of Neuroscience, University of Pennsylvania, Philadelphia, PA 19104, USA. [6] Department of Neurobiology, Neurobiology graduate program, Duke University Medical Center, Durham, NC 27710, USA. [7] Institute of Global Innovation Research, Tokyo University of Agriculture and Technology, Koganei, Tokyo 184-8588, Japan. [8] Duke Institute for Brain Sciences, Duke University, Durham, NC 27710, USA. [9] Present address: Department of Radiation Oncology, Helen Diller Comprehensive Cancer Center, University of California San Francisco, San Francisco, CA, USA. These authors contributed equally: Hitoshi Kida, Yosuke Fukutani, Joel D. Mainland. Correspondence and requests for materials should be addressed to H.M. (email: hiroaki.matsuami@duke.edu)

Animals have developed a highly complex olfactory system for detecting and discriminating a myriad of volatile odorous molecules, or odorants. In mammals, odorants entering the nose dissolve in the nasal mucus, which contains several metabolic enzymes capable of modifying odorants before binding olfactory receptors (ORs). Each odorant activates a specific set of ORs, which compose the largest family of G protein-coupled receptor superfamily with ~400 functional members in humans and ~1100 members in mice, individually expressed in olfactory sensory neurons (OSNs)[1–8]. Activated ORs couple with the G-proteins resulting in an increase in cAMP to ultimately lead to OSN depolarization and action potential firing. Second order neurons in the olfactory bulb receiving input from OSNs project to olfactory cortex to ultimately lead to the formation of odor perception[9,10].

Taking advantage of the highly sensitive and discriminative nature of the olfactory system, trained animals, such as dogs, have been used for detecting specific targets, including those relevant to disease diagnosis, environmental toxins, drugs, and terror agents[11,12]. Since then, biomimetic "noses", or devices to detect and discriminate target volatiles using OR proteins, have been proposed to replace trained animals. Previous studies using well-characterized model ORs have shown that various platforms can detect target odorants with high sensitivities[13–16], raising the possibility that multiplexed OR-based sensors using specific receptors for targeted odorants would have high sensitivity and discriminability.

Here, we conducted a large screening to identify ORs that robustly respond to seven target odorants. Subsequently, we developed a system to detect and discriminate volatile odorants in real time using a subset of ORs expressed in heterologous cells. We show that our set of 31 ORs are capable of discriminating small differences in chemical structures. Additionally, we demonstrate that a carboxyl esterase expressed in the olfactory mucus has odorant- and OR- specific roles in modifying OR activation in heterologous cells.

## Results

**Identification of a set of odorant-OR pairs**. To identify ORs that robustly respond to odorants in vitro, we conducted a large-scale screen of mouse ORs against a panel of seven odorants: acetophenone, cyclohexanone, eugenol, heptanal, 2-heptanone, methyl benzoate and N-amyl acetate (Fig. 1a). These odorants, representing diverse functional groups (ester, ketone, allyl benzene and aldehyde) and structures (straight and cyclic aliphatic, and aromatic), are broadly used in the field[17–26]. Further, acetophenone, eugenol and heptanal have well-established ORs, Olfr160, also known as M72 for acetophenone[27], Olfr73, also known as mOR-EG for eugenol[28] and Olfr2, also known as I7 for heptanal[29,30]. To measure the response of the ORs, we leveraged the luciferase reporter gene assay that we established previously[31,32] (Fig. 1b).

In the primary screening, we tested a total of 813 mouse ORs with 100 μM of each odorant directly dissolved in media. From the tested 813 ORs, we selected 279 odorant-receptor pairs, based on positive responses, for further analysis in which we examined responses to at 1, 10, and 100 μM. This further screening resulted in 176 odorant-receptor pairs with 138 unique ORs yielding responses greater than no-odor controls (t-test, $p < 0.05$, uncorrected; Fig. 1c, d). The response profiles are ranked based on the fold change in luminescence and are reported in Supplementary Fig. 1 and Supplementary Data 1.

**Real time monitoring of OR activation by liquid stimulation.** To evaluate real-time responses of a panel of ORs in a single 96-well plate based assay, we selected 29 robustly responding ORs from our screen (Supplementary Fig. 1), including two previously well-studied receptors, Olfr2 (I7)[29] and Olfr124 (SR1)[33]. In addition, we added the two previously well-studied receptors Olfr160 (M72)[34] and Olfr73 (mOR-EG)[28] for a total of 31 ORs. The 31 ORs and a control vector (pCI) were individually transfected in triplicate in 96-well plates.

We first examined the response of the 31 ORs expressed in Hana3A cells with the tested odorants dissolved into the medium (Fig. 2a). OR-mediated responses were measured using the GloSensor system allowing us to monitor cAMP levels in real time. A normalized response profile of Olfr145 to 50 μM acetophenone is shown in Fig. 2b as an example. The data show increasing luminescence over time, an indication of OR activation by odor. To quantify OR activations, we analyzed the area under the curve (AUC) of the normalized luminescence (Fig. 2c). Color-coded representation of the AUC of each of the tested 31 ORs and a negative control shows the relative response of each OR to 50 μM of each of seven odorants (Fig. 2d, Supplementary Fig. 2 and Supplementary Data 2).

**OR activation via vapor-phase odor stimulation.** To better reflect OR activation in the nasal mucosa, we next examined the response of the 31 different ORs by stimulation with odorants via the vapor phase. The luminescence-measuring chamber of the plate reader was equilibrated with the tested odorants without directly mixing the tested odorant with the stimulation medium (see methods for details). In this assay, as in the mammalian olfactory system, the volatile odorant dissolves in the liquid medium surrounding the receptor before binding and activating the ORs (Fig. 3a). Estimated vapor concentrations of undiluted odorants in ppm for each odorant are shown in Supplementary Table 1.

A normalized response profile of Olfr145 to acetophenone diluted at $10^{-2}$ (vol/vol) in mineral oil is shown in Fig. 3b as an example. To delineate the time course of the response onset, we conducted an experiment with more frequent measurements using Olfr145 (Fig. 3c). The response started increasing the luminescence within 60 s (F (15,32) > 1.9, $p < 0.05$ one-way ANOVA followed by Dunnett's test) (Supplementary Table 2).

To rule out whether odorants which penetrate the plastic walls of the cell culture plates activate the tested ORs, we stimulated the cells with only pre-incubation of acetophenone in another plate in the plate reader prior to measurements, thus the only way to activate the ORs is through dissolving into the medium from the air. We observed dose-dependent activation of Olfr145 by acetophenone, confirming that odorants in vapor are able to activate the tested ORs (Fig. 3d).

Comparing the response of Olfr145 by acetophenone in liquid versus vapor phase stimulation using the GloSensor approach, revealed a significant linear relationship between 50 μM liquid phase stimulation and $10^{-2}$ vapor phase stimulation ($R^2 = 0.89$) (Fig. 3e left). Comparison of 50 μM liquid phase stimulation to $10^{-4}$ vapor phase stimulation also revealed a significant but weaker relationship ($R^2 = 0.69$) (Fig. 3e right), likely due to no/little response for most ORs at $10^{-4}$ acetophenone. Overall, we observed significant positive correlations for all the tested odorants with varying $R^2$ ($10^{-2}$ dilution: $p < 10^{-6}$. $R^2 = 0.52–0.89$) ($10^{-4}$ dilution: $p < 2 \times 10^{-4}$. $R^2 = 0.36-0.92$) (Fig. 3f, Supplementary Figs. 3, 4). Altogether, these results demonstrate that our vapor stimulation assay is capable of monitoring OR activity by volatile odorants in real time.

**Activation profiles of the OR panel.** We then tested the 31 ORs with vapor stimulation at $10^0$ (undiluted), $10^{-2}$, $10^{-4}$, $10^{-6}$ and

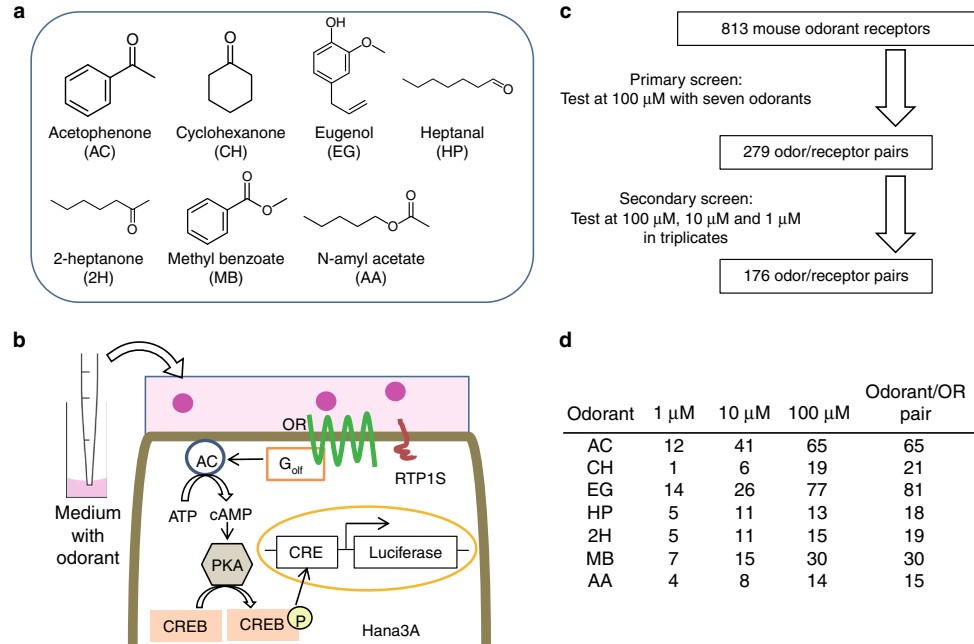

**Fig. 1** A large-scale screening to identify OR-odorant pairs. **a** Odorants tested in the assay. **b** Schematic representing the luciferase reporter gene assay in Hana3A showing OR signal transduction pathway. **c** Outline of the screening procedure. **d** The number of ORs that passed the secondary screen for each odorant ($t$-test, $p < 0.05$ uncorrected). AC adenylyl cyclase, ATP adenosine triphosphate, cAMP cyclic monophosphate, CRE cAMP response element, CREB cAMP response element binding protein, PKA protein kinase A, RTP1S receptor transporting protein 1 short

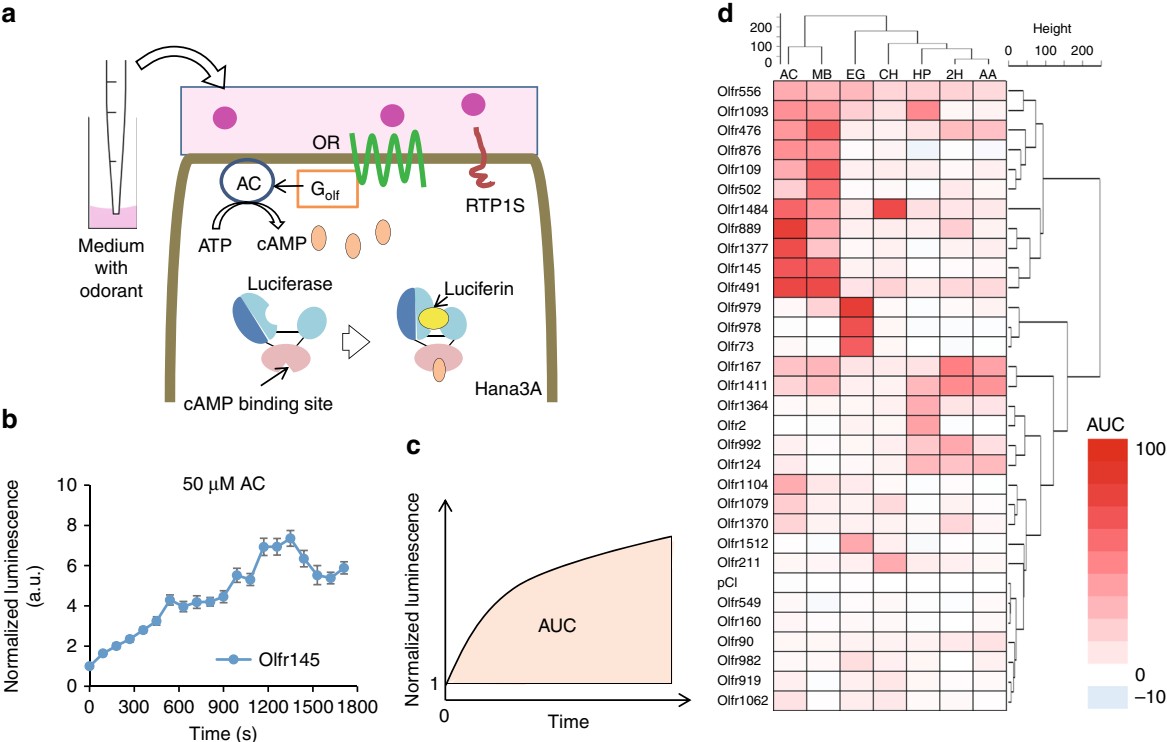

**Fig. 2** Liquid-phase odor stimulation with GloSensor assay system. **a** A schematic diagram of GloSensor assay in Hana3A showing OR signal transduction pathway. The GloSensor system use a modified luciferase with a cAMP binding domain so that luminescence activity depends on cAMP. **b** Real time measurement of each OR activation with 50 μM acetophenone. Error bars indicate s.e.m. ($n = 3$). **c** The image of area under the curve (AUC) analysis. Here, AUC indicates the sum of normalized luminescence from initial response to final response. **d** The heat map based on AUC with 50 μM acetophenone stimulation. The number on the left side indicate each OR and pCI (vector control) with a hierarchical clustering analysis. AC adenylyl cyclase, ATP adenosine triphosphate, cAMP cyclic monophosphate, RTP1S receptor transporting protein 1 short

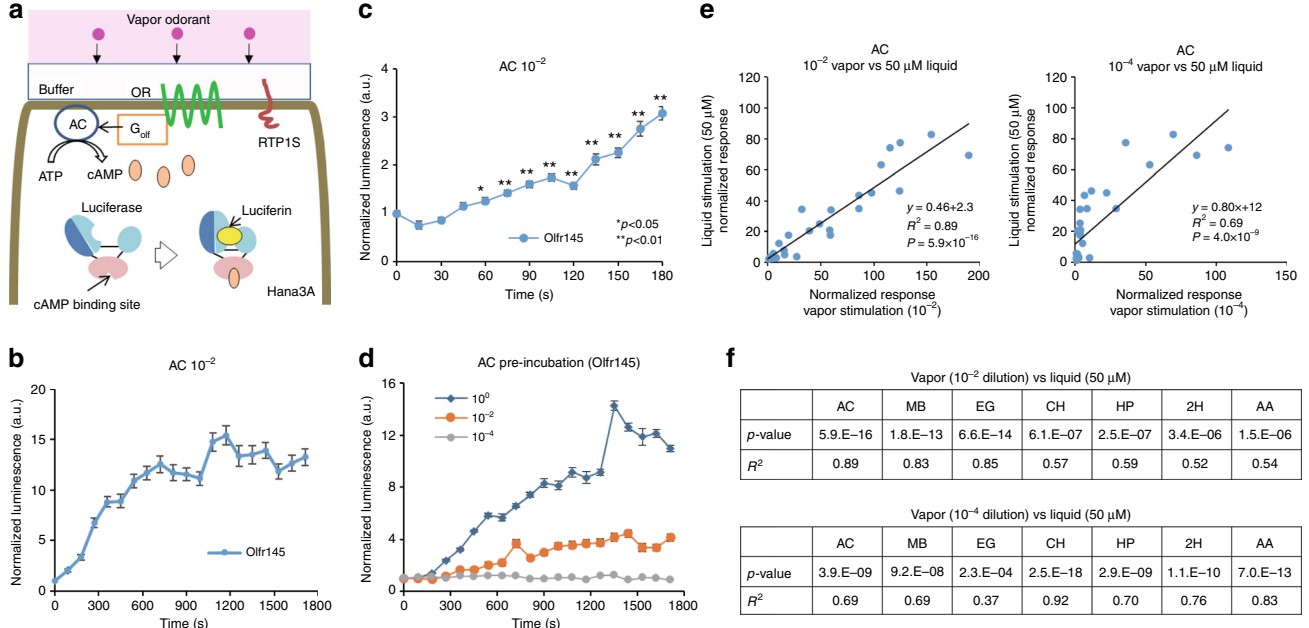

**Fig. 3** Vapor detection with GloSensor assay system. **a** Schematic representing the GloSensor assay in Hana3A showing OR signal transduction pathway. **b** Real time measurement of each OR activation with $10^{-2}$ dilution of acetophenone (AC). Luminescence in each well was measured once every 90 s for 20 cycles and the values were normalized with the initial value for each receptor and response of the vector control at the given time point. Error bars indicate s.e.m. ($n = 3$). **c** More frequent (15 s intervals) measurements using Olfr145 and a vector control with $10^{-2}$ dilution of acetophenone. Error bars indicate s.e.m. ($n = 3$). **d** Olfr145 activation with only pre-incubation of the luminescence measuring chamber with acetophenone. Error bars indicate s.e.m. ($n = 3$). **e** Comparison of the normalized response by vapor and liquid stimulation. A $10^{-2}$ or $10^{-4}$ dilution of acetophenone in mineral oil was placed adjacent to the wells for vapor stimulation. Furthermore, 50 μM of acetophenone was used for liquid stimulation. The straight line is the regression line and $R^2$ indicates Pearson's correlation coefficient. The $p$-value was calculated from the regression analysis. **f** Summarize of comparison between vapor ($10^{-2}$ or $10^{-4}$ dilution) and liquid (50 μM) stimulation using seven odorants. AC adenylyl cyclase, ATP adenosine triphosphate, cAMP cyclic monophosphate, RTP1S receptor transporting protein 1 short

$10^{-8}$ (vol/vol) concentrations in mineral oil. Responses of the most sensitive OR for each of the tested odorants are shown in Fig. 4. Color-coded representations of the AUC of each of the tested ORs show the relative response of each OR to the seven odorants at varying concentrations (Fig. 5). Comparing the response profile at different concentrations, we observed, as expected, increasingly robust activation of the tested ORs with higher odor concentrations, up to $10^{-2}$. Likewise, ORs activated by lower concentrations were subsets of ORs activated at higher concentrations (Fig. 5, Supplementary Figs. 5–17 and Supplementary Data 3). Strongly activated ORs in our luciferase reporter gene assays (Supplementary Data 1) responded to the same odorant in the Glosensor assays.

At $10^0$ (undiluted) concentrations, we observed varying degrees of diminished responses. To test the possibility of high odorant concentrations influencing cell viability and toxicity, we conducted the CellTiter-Glo cell viability assay (see methods for details). Eugenol and methyl benzoate were chosen as representative odorants based on our observations that $10^0$ eugenol yielded a dramatic reduction in activity whereas $10^0$ methyl benzoate yielded only a moderate reduction in activity. A 75% reduction of cell viability was observed with $10^0$ eugenol vapor stimulation compared with no odor control after 2 h, but a much more moderate reduction of cell viability was observed with $10^0$ methyl benzoate (Supplementary Fig. 18). Altogether, these data are consistent with the idea that high odorant concentration influence cell viability and odorant induced responses.

Altogether the 31 ORs show differential activation among the panel of tested odorants (F (6,14) > 2.85, $p < 0.05$, one-way ANOVA). The 2–21 ORs were differentially activated when we compared each pair of the tested odorants ($p < 0.05$, Tukey's post hoc analysis) (Supplementary Data 4).

**Differential activation of ORs with similar structures**. We next asked whether the 31 OR panel could discriminate between structurally analogous odorants including odors that were not in the initial screen, with vapor phase stimulation at $10^{-2}$ odorant concentration. We began with acetophenone and six of its analogs: 4-methyl acetophenone, propiophenone, benzaldehyde, 2-hydroxy acetophenone, menthone, and methyl salicylate (Fig. 6a, b, Supplementary Fig. 19 and Supplementary Data 5).

The majority of the tested ORs (27/31) showed differential activation among the tested analogs (F (6,14) > 2.85, $p < 0.05$, one-way ANOVA). To visualize the 32-dimensional representation of the seven odors we used the $t$-Distributed Stochastic Neighbor Embedding ($t$-SNE)[35] dimensionality reduction technique (Fig. 6c). Furthermore, 5–20 ORs were differentially activated when odorants were compared by pair ($p < 0.05$, Tukey's post hoc analysis), indicating that each odorant showed a unique and discriminative response pattern among the OR panel (Fig. 6d and Supplementary Data 6). Further, we evaluated the reproducibility and discrimination potential of this assay. Linear regression analysis on the response of 31 ORs against the same odorant in independent experiments indicated the high reproducibly ($R^2 = 0.91$-$0.97$) (Supplementary Fig. 20). And as expected, correlation between the response of 31 ORs against different analogs in the same experiment was significantly lower than against the same odorant in independent experiments ($R^2 = 0.14$-$0.85$, $p < 0.0001$, Student $t$-test) (Fig. 6e, f and Supplementary Fig. 21).

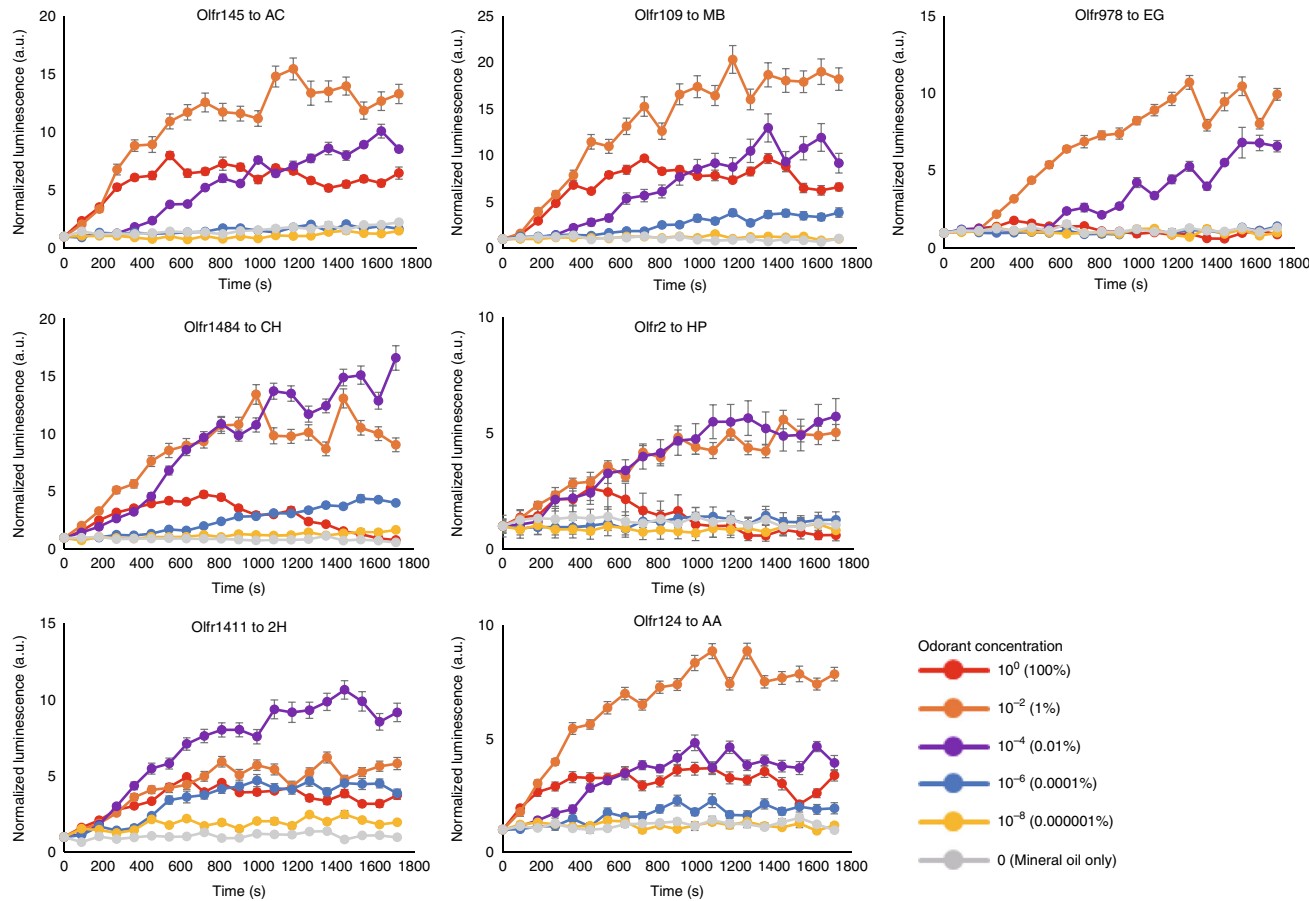

**Fig. 4** Real time measurement of sensitive ORs for each odorant. Combinations of OR and abbreviations for the odorants are as follows; Olfr1377 to acetophenone (AC), Olfr109 to methyl benzoate (MB), Olfr978 to Eugenol (EG), Olfr1484 to cyclohexanone (CH), Olfr2 to heptanal (HP), Olfr1411 to 2-heptanone (2H) and Olfr124 to N-Amyl acetate (AA). The luminescence in each well was measured once every 90 s for 20 cycles. The luminescence was normalized such that the initial value of each receptor was defined. Error bar indicated s.e.m. (n = 3)

Similar results were obtained when we tested eugenol and its analogs (2-methoxy-4-methyl phenol, guaiacol, methyl isoeugenol, methyl eugenol, eugenol acetate and ethyl vanillin) (Supplementary Figs. 22–25 and Supplementary Data 7).

To address whether the repertoire of 31 ORs can distinguish among these compounds, we trained a random forest classifier on assay responses for three replicates of seven odors tested against 31 ORs and a vector control. We then tested the same odors against the same ORs in triplicate on a different day and used the classifier to predict the odors presented in this independent dataset. The random forest classifier was 95.2% accurate, with the only error occurring when the classifier predicted benzyl aldehyde for receptor responses to 2-hydroxy acetophenone (Supplementary Data 8). Together, these results indicate that the OR panel is capable of discriminating between structural analogs.

**Alteration of OR responses by a metabolic enzyme.** It has been previously shown that members of the carboxyl esterase (Ces) enzyme family, known to metabolize carboxyl ester groups into alcohols and carboxylic acid, are expressed in the olfactory mucosa of mammals[36,37] and pharmacological inhibition of Ces results in changes in odor-mediated OSN activities[2]. Development of an OR-based volatile odorant sensor allowed us to functionally interrogate the role of xenobiotic enzymes like Ces1d, and its role in modulating specific OR response patterns. We focused on Ces1d, the most abundant Ces expressed in the olfactory mucosa and an ortholog of the human Ces3[38,39], as a model enzyme.

To examine the role of Ces1d in modulating OR response, we cloned Ces1d from mouse olfactory epithelium cDNA and co-expressed it with ORs in our vapor stimulation system. We analyzed the response of the OR panel co-transfected with Ces1d stimulated by three carboxylic esters: eugenol acetate, benzyl acetate and N-amyl acetate (Supplementary Fig. 26 A). OR responses displayed both enhancement and suppression with co-expression of Ces1d. Interestingly, these changes were specific to OR-odorant combinations (Fig. 7 and Supplementary Data 9). Consistent with the enzymatic action of Ces1d, Olfr979, which does not respond to eugenol acetate but eugenol, responded to eugenol acetate when co-expressed with Ces1d (2.8-fold change, FDR corrected $p < 0.01$) (Fig. 7a and Supplementary Fig. 26B). In contrast Ces1d co-expression suppressed the response of Olfr979 to benzyl acetate (0.13-fold change, FDR corrected $p < 0.01$) and to N-amyl acetate (0.41-fold change, FDR corrected $p < 0.01$) (Fig. 7c, e). In sum, Ces1d significantly changed activities of five ORs against Eugenol acetate, 15 ORs against Benzyl acetate and eight ORs against N-amyl acetate (FDR corrected $p < 0.05$) (Fig. 7b, d, f and Supplementary Data 10). These data suggest a complex odorant-specific role of Ces1d in modulating OR responses

## Discussion

In this study, we developed an OR-based sensor and utilized it to identify and discriminate between a panel of responding odorants in vitro. Furthermore, our OR sensor array demonstrated the

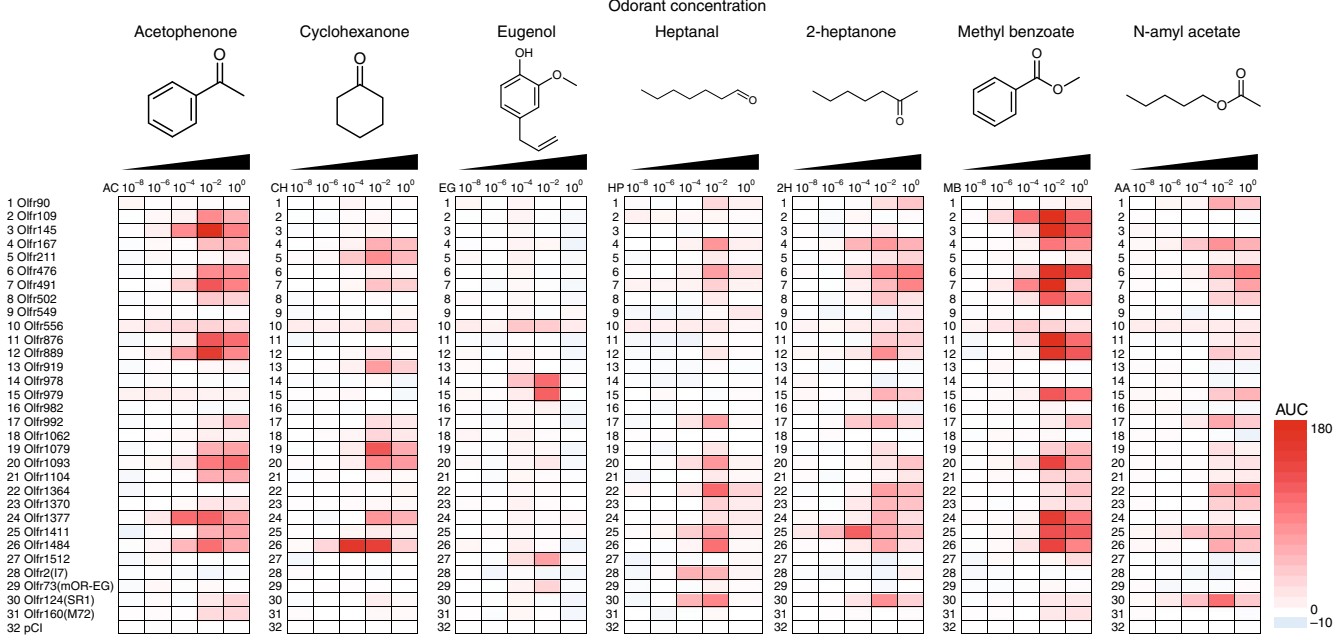

**Fig. 5** Heat map of AUC. The number on the left side indicate each OR as follows: 1; Olfr90, 2; Olfr109, 3; Olfr145, 4; Olfr167, 5; Olfr167, 6; Olfr211, 7; Olfr476, 8; Olfr491, 9; Olfr502, 10; Olfr549, 11; Olfr556, 12; Olfr876, 13; Olfr889, 14; Olfr978, 15; Olfr979, 16; Olfr982, 17; Olfr992, 18; Olfr1062, 19; Olfr1079, 20; Olfr1093, 21; Olfr1104, 22; Olfr1364, 23; Olfr1370, 24; Olfr1377, 25; Olfr1411, 26; Olfr1484, 27; Olfr1512, 28; Olfr2(I7), 29; Olfr73(mOR-EG), 30; Olfr124(SR1), 31; Olfr160(M72), 32; pCI (Vector control)

capability to detect and discriminate structurally related odorants delivered in vapor phase.

Volatile odorous molecules activate ORs after they dissolve in the nasal mucus and are possibly modified by mucosal enzymes in vivo. Widely used in vitro systems to measure OR activation have used odorants dissolved into the cell medium. To bridge the gap between the in vivo and in vitro and to realize a biomimetic volatile chemical sensor to rival animals' ability in detection and discrimination, we must overcome multiple challenges, including, reconstructing features of the nasal mucus, identifying best combinations of ORs for detection and discrimination of targeted odorants among structurally related chemicals, and monitoring OR activation patterns in real time. In this study, we made progress towards these goals on multiple fronts. First we conducted a large-scale screening to identify ORs responding robustly to odorants of interest, including methyl benzoate, the active odor of cocaine[40] and cyclohexanone, the odor component released from explosives[41,42]. Second, we developed a system that is capable of detection and discrimination of both structurally similar and diverse odorants presented in vapor phase. Third, using this approach we characterized the function of an olfactory mucosal enzyme modulating specific OR activities.

A given odorant activates multiple ORs, some of which respond more robustly than others. Previous OR-based sensors have used a small number of ORs that are not optimized to detect the tested odorants. Here, we conducted a large-scale cell-based screening using approximately 74% of the mouse OR repertoire to identify robustly responding ORs to a set of odorants.

The ORs that passed our screening are likely to be included among the most sensitive ORs activated by the odorants using the heterologous assay platform. This is supported by our analysis in comparison with well-characterized ORs. For example, Olfr160 (M72) was identified as an acetophenone receptor[27], but is not among the most sensitive or most robustly responding receptors for acetophenone in vivo[34,43,44]. Consistently, it shows only moderate activation in our assays (Fig. 5). Our screening identified many ORs that are strongly activated by acetophenone as

well as six additional odorants from which we selected 31 ORs in the subsequent work. Some of the selected ORs responded to multiple odorants with diverse structures while others responded to one of the seven target ORs. These are consistent with the known feature of overall OR repertoire; some ORs are broadly-tuned "generalist receptors", others are narrowly-tuned "specialist receptors"[45,46].

Previous biomimetic nose studies have provided limited knowledge in evaluating how much discriminability a given system has, since they used a small number of ORs (up to four) and primarily tested only known ligands[13,16,47].

In the olfactory system, odorants reach to the olfactory epithelium as volatiles and are dissolved in the olfactory mucus before activating the ORs. In typical heterologous cell-based assay systems, however, odor stimulation is performed by replacing the medium with odor-containing medium where odorants are dissolved[14,18,30,47–52]. Our new assay system aims to better mimic the olfactory system in the heterologous cells by monitoring real time activation events in the presence of volatile odor in the reading chamber. Only a small number of studies using insect or human ORs have measured activation by volatile odors[53,54].

Here, we monitored activation of a large array of 31 ORs that respond to volatile chemicals and demonstrated that the panel discriminates structurally related odorants. Comparing both acetophenone and eugenol analogs, at least one OR was differentially activated when against any two structurally similar odorants, suggesting that our system is highly selective. The seven odorants used to screen ORs is likely to bias the choice of the ORs, which in turn may limit its applicability as a general odor sensor. One limitation of our new assay system is the dynamic range is low for low-affinity ORs due to toxicity of the odorants at high concentrations.

Finally, future work to generate an OR activation database with increased odorants and concentrations should be useful as a shared resource for the community. Further, miniaturization of the system will enable us to include more ORs to enhance detect and discriminability.

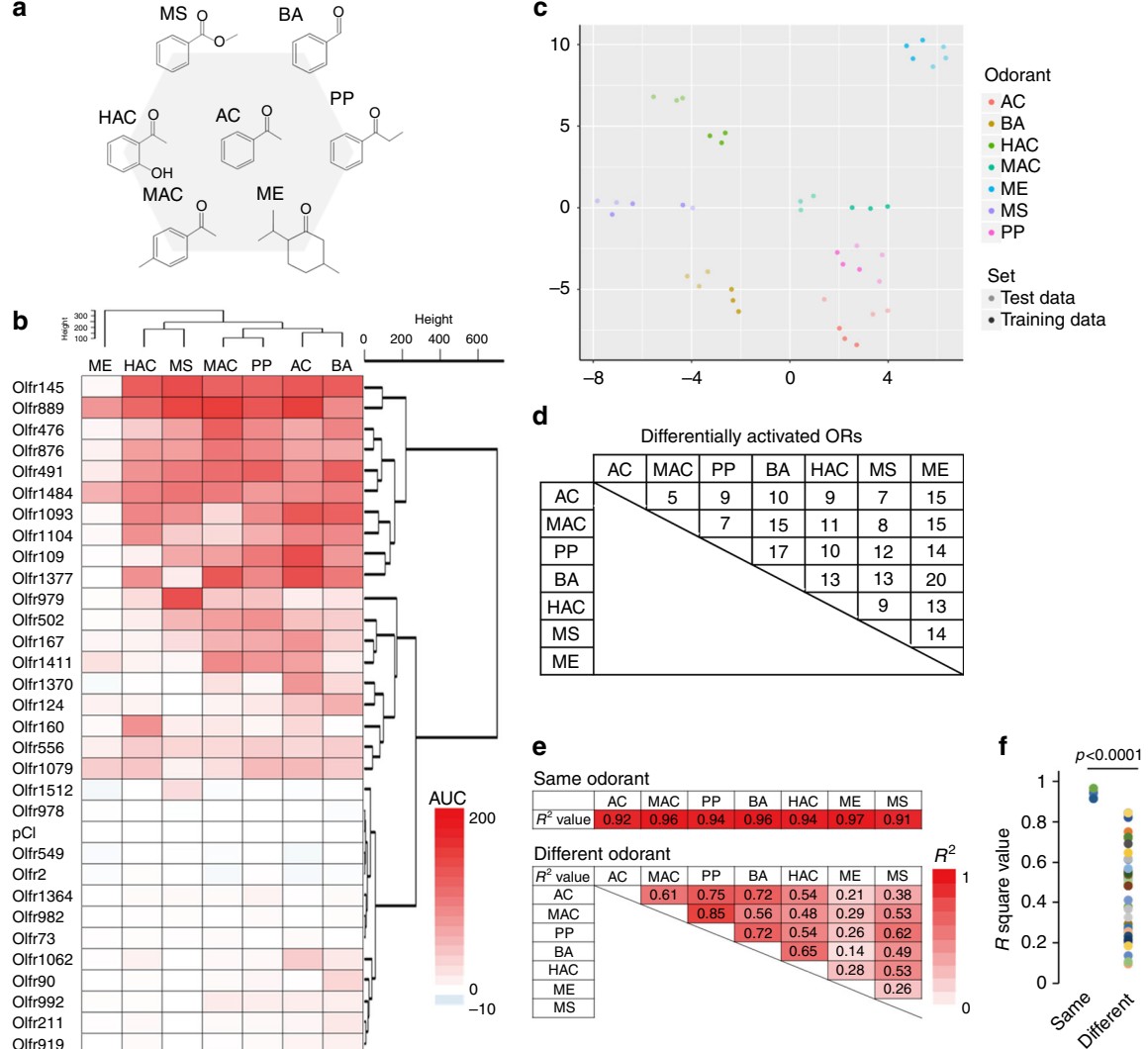

**Fig. 6** Differential activation of ORs with acetophenone analogs. **a** Chemical structures of each acetophenone analog used in this study. **b** Heat map showing AUC of each OR stimulated with odorants at $10^{-2}$ dilution. The number on the left side indicates each OR. **c** The 32-dimensional space (31 ORs and a vector control) visualized using t–SNE. Each point is an odor with color representing the odor presented and saturation representing whether the point was used to train the model or was part of the test set. **d** The number of ORs that are differentially activated by tested odorant (Tukey's post hoc analysis, $p < 0.05$). **e** Summary of the discrimination analysis using the same or different odorant pairs. Each $R^2$ value is shown in Supplementary Figures 20 and 21. **f** Difference of $R^2$ values between results with the same odorant in different experiments and different odorants in the same experiment Student's $t$-test. AC acetophenone, MAC 4-methyl acetophenone, BA benzaldehyde, PP propiophenone, HAC 2-hydroxyl acetophenone, MS methyl salicylate, ME menthone

Nasal mucus contains a variety of enzymes functioning in the peri-receptor events[55] to facilitate hydrophobic molecule dissolution, odorant transport, odorants metabolism, or toxin clearance. While we focused on Ces1d, a member of the carboxyl esterase enzyme family, the nasal mucus also has been shown to contain carboxyl esterase, cytochrome P450, aldehyde dehydrogenases, glutathione transferases, as well as others, altogether creating complex environment[2,56–58]. Our system may have a unique advantage in studying peri-receptor events in heterologous cells, because, unlike traditional heterologous system, odorants have to dissolve into the stimulation medium before activating ORs. Though the results should be interpreted with caution in relating to their in vivo function, our data clearly shows that Ces1d significantly affects OR activation in vitro. Our data suggests that the mode of action of Ces1d is complicated since it seems that the effects are both OR- and odorant- specific.

Some ORs showed an increase or decrease in response by co-expression of Ces1d. Future work is necessary to characterize substrate preferences of Ces1d, quantifying metabolites and assessing OR activations by each metabolite. Furthermore, it will be interesting to perform a large-scale screening in the presence of Ces1d. Nevertheless, our study underscores the importance of metabolic enzymes in affecting OR activation.

There are various components other than xenobiotic metabolizing enzymes in the nasal mucosa. Especially odorant binding proteins, which belong to lipocalin superfamily, are considered to play an important role in odor detection. Each of them may bind with a specific subset of odorants, which may help dissolution or transport of odorants and modulate OR activation patterns[59,60]. In conclusion, our expression system is likely to serve as a powerful tool in future studies on peri-receptor events that modulate OR activation.

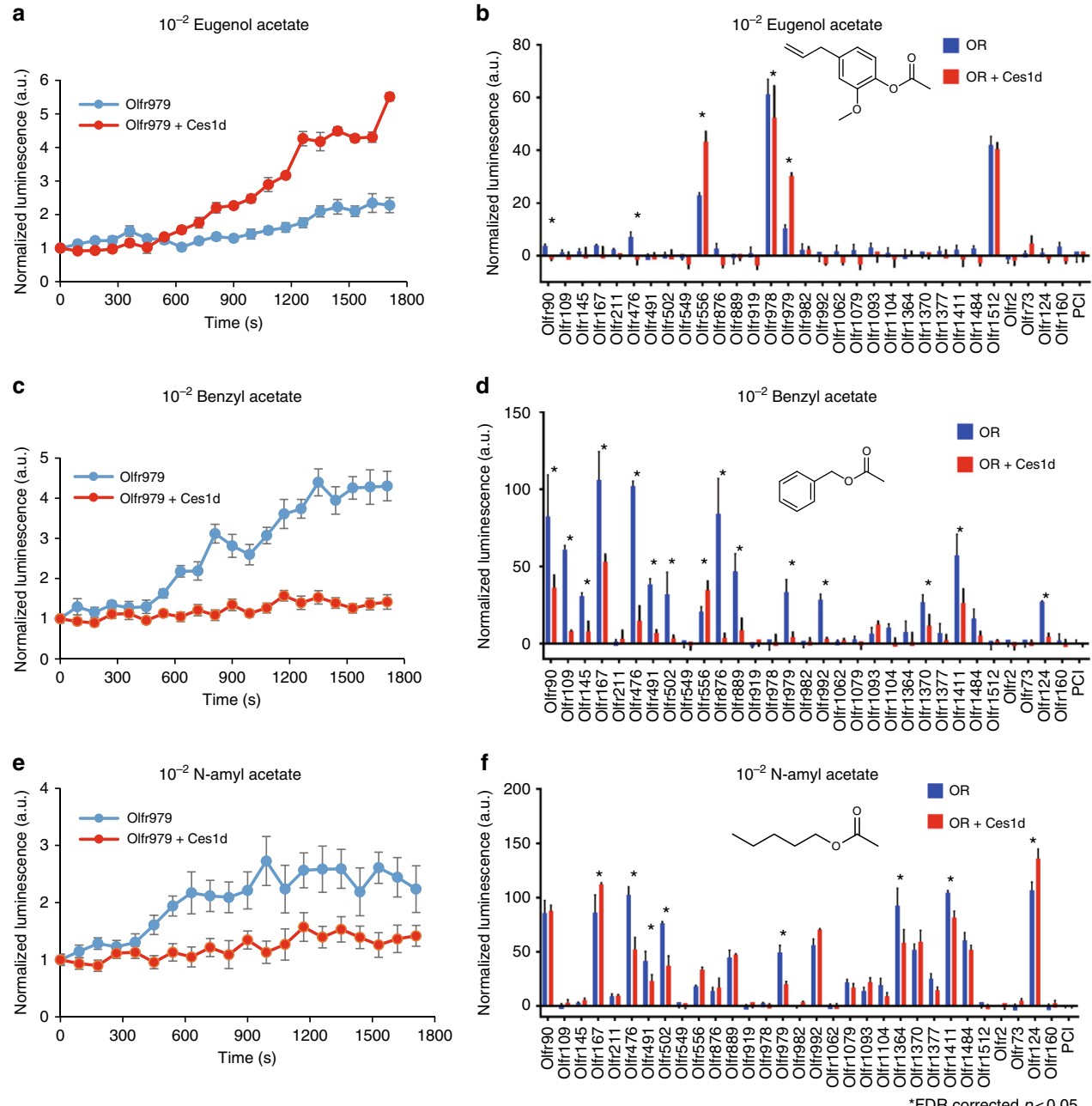

**Fig. 7** Effect of co-expression of Ces1d with ORs. Real time measurement of Olfr979 to **a** Eugenol acetate, **c** Benzyl acetate and **e** N-amyl acetate. Ces1d enhanced activation of Olfr979 to Eugenol acetate, but suppressed the activation of Olfr491 to benzyl acetate and N-amyl acetate. Error bar indicates s.e.m. ($n = 3$). **b-f** Identification of OR and odorant pair that change the activation with 1% odorant by Ces1d co-expression. The bars show that each normalized AUC value of OR only (red) and Ces1d (blue). Error bars indicate s.e.m. ($n = 3$). Statistical significance was assessed with FDR correction (*$p < 0.05$)

## Methods

**DNA and vector preparation**. The open reading frames of ORs were amplified using Phusion polymerase (Thermo Fisher Scientific). Amplified fragments were cloned into pCI expression vector (Promega) containing the sequence encoding the first 20 amino acids of human rhodopsin (Rho-tag) at the N-terminal[30]. The cDNA of Ces1d was amplified using Phusion polymerase from the cDNA library of mouse olfactory epithelium and cloned into pCI expression vector without any tag. The sequences of the cloned receptors were verified by sequencing (3100 Genetic Analyzer, Applied Biosystems).

**Cell culture**. Hana3A cells[61] were maintained in minimal essential medium (MEM) containing 10% FBS (vol/vol) with penicillin-streptomycin and amphotericin B at 37 °C and 5% $CO_2$. These cells were authenticated using polymorphic short tandem repeat (STR) at the Duke DNA Analysis Facility using GenePrint 10 (Promega) and shown to share profiles with the reference (ATCC). No mycoplasma infection was detected.

**Luciferase reporter gene assay**. In the large-scale screening, the Dual-Glo Luciferase Assay System (Promega) was used to measure receptor responses as previously described[31]. Hana3A cells were plated on 96-well plates. Approximately 18–24 h after plating, cells were transfected with 5 ng/well of plasmids encoding OR, 2.5 ng/well of M3-R, 5 ng/well of RTP1S, 10 ng/well of CRE-luciferase and 5 ng/well of pRL-SV40. Furthermore, 18–24 h later, cells were stimulated by incubation with an odorant diluted in CD293 (Gibco) at 37 °C and 5% $CO_2$ to allow for CRE-luciferase expression. Four hours after stimulation, luminescence was measured a Polarstar Optima plate reader (BMG). All luminescence values were divided by *Renilla* luciferase activity to control for transfection efficiency in a given well. Each comparison was performed in triplicate. Statistical significance was assessed by a two-sided *t*-test comparing the wells stimulated with odor with the three wells stimulated with medium alone.

**Vapor detection**. In the vapor odorant detection test, Glosensor cAMP Assay (Promega) was used to measure the real time changes in cAMP levels caused by

receptor activation upon ligand binding. Hana3A cells were plated on 96-well plates. 18–24 h after plating, cells were transfected with 80 ng/well of plasmids encoding ORs, 5 ng/well of RTP1S and 10 ng/well of Glosensor plasmid (Promega). Then, 18–24 h later, the medium was replaced with 25 µL of HBSS (Gibco) containing 10 mM HEPES and 1 mM Glucose, followed by 25 µL of the HBSS containing GloSensor cAMP Reagent (Promega). Plates were kept in a dark place at room temperature for two hours to equilibrate cells with the reagent. Odorant chemicals were diluted in Mineral oil (Sigma-Aldrich). Before odor stimulation of the cells expressing individual ORs on testing 96-well plate by odorants, a 96-well plate filled with 25 µL of tested odorant with corresponding dilution in mineral oil and placed in the plate reader for 5 min to equilibrate the reading chamber. Just before replacing the plate with cells, we filled the gaps between the wells of the plates with the mineral oil containing odorant at the same concentration, allowing maintaining an equilibrium of the tested odorant without direct contact with the cells. Immediately, the test plate was inserted in the plate reader. The luminescence in each well was measured at 90 s intervals for 20 cycles. All luminescence values were divided by the value obtained from the cells transfected with the empty vector at the same cycle. After measuring, the remaining volatiles inside the reading chamber was extensively vacuumed and replaced with fresh air. We obtained very similar results in OR responses using AUC values and peak responses ($R^2 > 0.970$ at $10^{-2}$) (Supplementary Fig. 27 and Supplementary Data 11). Well-to-well variations tended to be lower when AUC values were used for analysis. Each comparison was performed in triplicate. Multiple comparisons were performed using one-way analysis of variance (ANOVA). Then we evaluated the probability that the means of two populations were equal using a Tukey's post hoc analysis. To evaluate the effect of Ces1d, we used the original FDR method of Benjamini and Hochberg.

**Cell viability assays**. To assess cell viability, Hana3A cells were plated at a 100% confluence in 96-well plates overnight. Cell viability was tested at $t = 0$ and after 2 h incubation in the luminometer chamber at room temperature in three different conditions; no odor, 100% methyl benzoate and 100% eugenol (odorant stimulation in vapor phase). The ATP content, assessing the cell viability, was monitored using CellTiter-Glo assay (Promega). Before odor treatment, the culturing media was replaced by 25 µL of HBSS containing 10 mM of HEPES and 1 mM of D-Glucose. After odor treatment, 25 µL of CellTiter-Glo Reagent were added to each well and the plate was incubated for 2 min with shaking and stabilized at room temperature for 10 min. Cell viability was assessed by measuring luminescence. Results after 2 h incubation were normalized to the $t = 0$ value. Multiple comparisons were performed using one-way analysis of variance (ANOVA) followed by Turkey multiple comparison test.

**Discriminating odors computationally**. 31 ORs and a vector control were tested against seven odorants. To visualize this 32-dimensional representation of the seven odorants we used the dimensionality reduction technique t-Distributed Stochastic Neighbor Embedding (t-SNE) with the perplexity parameter set to 10. To formally test the ability of the assay data to predict the odorant, we used a random forest classifier[62]. In a random forest, multiple decision trees are built from a random sampling of data with replacement (bootstrap samples). Furthermore, a random set of features are used to determine the best split at each node during the construction of a tree. Similar results were obtained using a linear discriminant analysis, with a single classification error (MAC was predicted for an AC trial). All models were implemented in the R statistical package version 3.5.0[63].

## Data availability

All relevant data are available within the manuscript and its supplementary information or from the authors upon reasonable request.

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

## Acknowledgements

We thank Mengjue Jessica Ni for expert technical assistance. We thank Marcelo Zapata and Yueyang Eric Lu for reading and editing the manuscript. This work was supported by grants from NIH (DC014423 and DC016224) and the Defense Advanced Research Project Agency RealNose Project. H.K. stayed at Duke University with financial support from Tokyo University of Agriculture and Technology as a student of the program for leading graduate schools in Japan. Y.F. stayed at Duke University with financial support from JSPS Program for Advancing Strategic International Networks to Accelerate the Circulation of Talented Researchers (R2801).

## Author contributions

H.K., J.D.M. and H.M. conceived and designed the project. J.D.M., Y.R.L., Q.C., L.L., A.T. and H.M. performed research concerning the large scale screening. H.K., Y.F., A.V. and C.D.M. performed research concerning the vapor detection system. H.K., Y.F., J.D.M. and H.M. carried out the analysis and wrote the paper with inputs from all authors. M.K., M.Y. and H.M. supervised the project.

## Additional information

**Competing interests:** H.K., Y.F. and H.M. filed a patent application relevant to this work on 27 October 2016. J.M. receives research funding from both Ajinomoto Co., Inc. and Proctor & Gamble. He is on the scientific advisory board of Aromyx and receives compensation for these activities. The remaining authors declare no competing interests.

