## [Peer Review File · Nature Communications]

Reviewers' Comments:

Reviewer #1:

Remarks to the Author:

This manuscript describes the development of an OR-based sensor, which was utilized to identify and discriminate odorants in vapor phase as well as in liquid phase. This article provides extensive and useful information about the combinatorial pattern and relationship between ORs and odorants. I recommend this article to be published in Nature Communications after minor revision.

1. Authors selected seven odorants: acetophenone, cyclohexanone, eugenol, heptanal, 2-heptanone, methyl benzoate and N-amyl acetate. They we conducted a large-scale screening of mouse ORs against these odorants. Although they described that these odorants represent diverse functional groups, sizes, and structures, it is not clearly. Authors need to describe more adequately what the rationale is for the selection of these seven chemicals as representative odorants.
2. The responses of ORs were diminished for all odorants at high concentrations of odorants, especially in undiluted cases. It was described that they are presumably resulting from toxicity on cells. Probably it is the case. Then, authors need to conduct the toxicity test in order to clarify how high concentration affect the cell viability or luciferase assay.
3. Methods for the vapor phase experiments are not so adequately described to be understood. More detailed procedures should be described in Materials and Methods.
4. This article also demonstrated that a metabolic enzyme expressed in olfactory mucosa affected the specificity between ORs and odorants. This kind of approach is meaningful in order to mimic the mammalian olfactory system more realistically. However, there are various components other than carboxyl esterase in the mucosa. Especially odorant binding proteins are considered to play an important role for smell sensing. Therefore, I suggest that this part should not be too strongly emphasized. Length of the description about this part should be also shortened.

Reviewer #2:

Remarks to the Author:

Matsunami's group has been studying interactions of odorant receptors (ORs) and their ligands using a heterologous assay system that they developed. In the present study, the authors performed a large-scale screening to identify ORs that respond robustly to odorants of interest. They also developed a heterologous system capable of discriminating structurally similar odorants in vapor. The authors further characterized an olfactory mucosal enzyme that modulates OR activities. These studies provide us with useful information for our understanding of odor detection and discrimination. Unfortunately, however, this reviewer finds that the paper needs more conceptual advancement to attract the broad audience of Nat. Commun. Thus, I would recommend this paper be published in a more specialized journal.

Specific comments are as follows:

1. In the text, it is not always clear which figure or table is being referred
2. In this study, the authors chose to use seven specific odorants in the first screening. Why were they selected? What was the basis for this selection?
3. Some main figures, e.g., Fig. 1, do not contain sufficient data and are suitable for supplementary figures.
4. The authors should emphasize what is novel in the present system compared with the previous luciferase assay.
5. In lines 155-157, the authors describe the toxicity of odorants at high concentrations on Hana cells. What is the evidence for this?
6. In lines 174-176, the authors emphasize the reproducibility of the present assay. The reproducibility should be compared between the present assay system and the previous one.
7. As for the Ces 1d enzyme, how does it work on odor ligands? Which part of ligand molecule is

digested?

8. This reviewer feels that the Ces 1d treatment should be performed in the first large-scale screening, rather than in the narrowed down OR set.

9. In lines 280-283, the authors claim that their new assay system directly detects vapor odorants. However, even in this assay, ORs expressed in Hana cells are detecting odorants dissolved in GloSensor cAMP reagent.

Reviewer #3:

Remarks to the Author:

Elucidating the chemical specificities of odorant receptors (ORs) as a group is critical for understanding the coding logic of olfaction. Such knowledge can also be applied to develop so-called biomimetic sensors that can be used as chemical detectors in a broad set of applications. However both goals require the ability to assay the ligand activation profiles of hundreds if not thousands of ORs in a context that recapitulates these receptors' function in vivo, namely to detect volatile chemicals. Thus, in this study Matsunami and colleagues describe a cell-based assay for odorant OR activation in response to odor ligands presented in the vapor phase. The authors initially screened over 800 mammalian ORs against 7 odors in liquid phase in a cell-based assay that reports on receptor activation of cAMP via a CREB-dependent transcription of a luciferase reporter gene. From this screen they winnowed the ORs down to 138 unique receptors showing statistically significant activity over background for further study. Next, the authors used a real-time assay system based on cAMP activation of an engineered luciferase whose activity is dependent on cAMP binding. The assay was then adapted to screen compounds in vapor phase and characterize the specificities of 32 selected ORs. From these analyses the authors argue that their assay is capable of discriminating both structurally divergent as well as similar odor molecules based on the differential activation of ORs within the set.

Over all, this study represents a technical tour de force and may indeed provide the foundation for building a biosensor for volatile chemicals. However in its current form the manuscript falls a bit short in its claims. My major concern is with the interpretation of the data in Figure 6 on the differential activation of ORs by structural variants of acetophenone. While the authors argue based on their statistical analysis that the repertoire of 32 ORs can distinguish among these compounds, setting aside menthone (ME) as an outlier, a qualitative assessment of the first 10 or so ORs in panel B might lead one to conclude that the responses are more or less the same. If the authors wish to make the argument that their assay can in principle serve as a biosensor capable of discriminating related compounds, they should put this to the test by e.g. using a subset of data as a training set to build a classifier, and then asking whether the classifier can reliably assign the identities of test odors in independent experiments.

Minor points:

The authors choose to use area under the curve (as somewhat unfortunate choice of terms, as AUC is typically used in ROC analyses) for the real-time assays to define the level of receptor activation. What is the rationale for doing so vs. using peak response?

The dynamic range of the real-time luciferase assay seems to be quite low, precluding the ability to generate any meaningful dose-response curve that should be expected to commence and saturate over 2 logs as expected for a pseudo-first-order reaction. While at face value the assay can discriminate compounds, the authors should comment on this limitation.

In Figure 3, the authors wish to demonstrate that OR activation is comparable whether the compounds are delivered in liquid or vapor phase. While the left-hand plot in panel E does convincingly show a linear relationship, the right-hand plot most decidedly does not; rather it is

biphasic or at least show a significant offset during the initial phase of the response. The authors should comment and perhaps temper their claims accordingly.

Responses to Reviewers:

Reviewer #1 (Remarks to the Author):

This manuscript describes the development of an OR-based sensor, which was utilized to identify and discriminate odorants in vapor phase as well as in liquid phase. This article provides extensive and useful information about the combinatorial pattern and relationship between ORs and odorants. I recommend this article to be published in Nature Communications after minor revision.

1. Authors selected seven odorants: acetophenone, cyclohexanone, eugenol, heptanal, 2-heptanone, methyl benzoate and N-amyl acetate. They we conducted a large-scale screening of mouse ORs against these odorants. Although they described that these odorants represent diverse functional groups, sizes, and structures, it is not clearly. Authors need to describe more adequately what the rationale is for the selection of these seven chemicals as representative odorants.

Response: We thank the reviewer for this valuable comment. We have revised the text to expand our rationale in Results and Discussion.

In Results, we added the following rationale on page 6 (changes are marked in red).

To identify ORs that robustly respond to odorants *in vitro*, we conducted a large-scale screen of mouse ORs against a panel of seven odorants: acetophenone, cyclohexanone, eugenol, heptanal, 2-heptanone, methyl benzoate and N-amyl acetate (Figure 1A). These odorants, representing diverse functional groups (ester, ketone, allyl benzene and aldehyde) and structures (straight and cyclic aliphatic, and aromatic), are broadly used in the field^{1, 2, 3 4 5, 6, 7, 8, 9, 10}. Further, acetophenone, eugenol and heptanal have well-established cognate ORs, Olfr160, also known as M72 for acetophenone¹¹, Olfr73, also known as mOR-EG for eugenol¹² and Olfr2, also known as I7 for heptanal^{13, 14}.

In Discussion, we added a sentence that strengthens rationale on page 13

First we conducted a large-scale screening to identify ORs responding robustly to odorants of interest, including methyl benzoate, the active odor of cocaine¹⁵ and cyclohexanone, the odor component released from explosives^{16, 17}

2. The responses of ORs were diminished for all odorants at high concentrations of odorants, especially in undiluted cases. It was described that they are presumably resulting from toxicity on cells. Probably it is the case. Then, authors need to conduct the toxicity test in order to clarify how high concentration affect the cell viability or luciferase assay.

Response: We thank the reviewer for the valuable suggestion. We have conducted additional experiments and analysis to evaluate the cell viability and observed a dramatic reduction of cell viability with 100% eugenol vapor suggesting high toxicity. In contrast, the majority of the cells survived with 100% methyl benzoate vapor stimulation, suggesting lower toxicity.

We revised the text to expand the Results on page 9 as follow (changes are marked in red).

At 10^0 (undiluted) concentrations, we observed varying degrees of diminished responses. To test the possibility of high odorant concentrations influencing cell viability and toxicity, we conducted the CellTiter-Glo cell viability assay (see methods for details). Eugenol and methyl benzoate were chosen as representative odorants based on our observations that 10^0 eugenol yielded a dramatic reduction in activity whereas 10^0 methyl benzoate yielded only a moderate reduction in activity. A 75% reduction of cell viability was observed with 10^0 eugenol vapor stimulation compared with no odor control after two hours, but a much more moderate reduction of cell viability was observed with 10^0 methyl benzoate (Supplementary Fig. 18). Altogether, these data are consistent with the idea that high odorant concentration influence cell viability and odorant induced responses.

Supplementary Fig. 18
Cell toxicity of odorants exposed to the cells using CellTiter-Glo® Luminescent Cell Viability Assay (Promega).

The luminescence in each well was measured at 120 min after stimulation by vapor phase odorant. The values of luminescence were normalized such that the value of each well on before stimulation was defined 1. Error bars indicate s.e.m. (n=4). The p-value was calculated with one-way ANOVA followed by Turkey multiple comparison test (**p<0.01, ****p<0.0001).

We also added its procedure in the Material and Method section on page 19 as follows

Cell viability assays

To assess cell viability, Hana3A cells were plated at a 100% confluence in 96-well plates overnight. Cell viability was tested at t=0 and after 2h incubation in the luminometer chamber at room temperature in three different conditions; no odor, 100% methyl benzoate and 100% eugenol (odorant stimulation in vapor phase). The ATP content, assessing the cell viability, was monitored using CellTiter-Glo assay (Promega). Before odor treatment, the culturing media was replaced by 25 μ l of HBSS containing 10mM of HEPES and 1mM of D-Glucose. After odor treatment, 25 μ l of CellTiter-Glo Reagent were added to each well and the plate was incubated for 2min with shaking and stabilized at room temperature for 10min. Cell viability was assessed by measuring luminescence. Results after 2h incubation were normalized to the t=0 value. Multiple comparisons were performed using one-way analysis of variance (ANOVA) followed by Turkey multiple comparison test.

3. Methods for the vapor phase experiments are not so adequately described to be understood. More detailed procedures should be described in Materials and Methods.

Response: Following the reviewer's suggestion, we described the procedure of the vapor stimulation assay in more detail. We revised the Material and Method section on page 18 as follows (changes are marked in red):

Vapor detection

In the vapor odorant detection test, Glosensor cAMP Assay (Promega) was used to measure the real time changes in cAMP levels caused by receptor activation upon ligand binding. Hana3A cells were plated on 96-well plate. 18-24 hours after plating, cells were transfected with 80 ng/well of plasmids encoding ORs, 5ng/well of RTP1S and 10ng/well of Glosensor plasmid (Promega). 18-24 hours later, the medium was replaced with 25 μ L of HBSS (Gibco) containing 10mM HEPES and 1mM Glucose, followed by 25 μ L of HBSS containing GloSensor cAMP Reagent (Promega). Plates were kept in dark place at room temperature for two hours to equilibrate cells with the reagent. Odorants were diluted in Mineral oil (Sigma-Aldrich). Before odor stimulation of the cells expressing individual ORs on testing 96 well plate by odorants, a 96 well plate filled with 25 μ L of the tested odorant with corresponding dilution in mineral oil and placed in the plate reader for 5 minutes to equilibrate the reading chamber. Just before placing

the plate with cells, we filled the gaps between the wells of the plate with mineral oil containing odorant at the appropriate dilution, allowing to maintain an equilibrium of the tested odorant without direct contact with the cells. Immediately, the test plate was inserted in the plate reader. The luminescence in each well was measured at 90 sec intervals for 20 cycles. All luminescence values were divided by the value obtained from the cells transfected with the empty vector at the same cycle. After measuring, the remaining volatiles inside the reading chamber was extensively vacuumed and replaced with fresh air. We obtained very similar results in OR responses using AUC values and peak responses ($R^2 > 0.970$ at 10^{-2}) (Supplementary Fig. 19 and Supplementary Data 11). Well-to-well variations tended to be less when AUC values were used for analysis. Each comparison was performed in triplicate. Multiple comparisons were performed using one-way analysis of variance (ANOVA). Then we evaluated the probability that the means of two populations were equal using a Tukey's post hoc analysis. To evaluate the effect of Ces1d, we used the FDR method of Benjamini and Hochberg.

4. This article also demonstrated that a metabolic enzyme expressed in olfactory mucosa affected the specificity between ORs and odorants. This kind of approach is meaningful in order to mimic the mammalian olfactory system more realistically. However, there are various components other than carboxyl esterase in the mucosa. Especially odorant binding proteins are considered to play an important role for smell sensing. Therefore, I suggest that this part should not be too strongly emphasized. Length of the description about this part should be also shortened.

Response: Following the reviewer's suggestion, we have shortened this part on page 11 as follows.

It has been previously shown that members of the carboxyl esterase (Ces) enzyme family, known to metabolize carboxyl ester groups into alcohols and carboxylic acid, are expressed in the olfactory mucosa of mammals^{18, 19} and pharmacological inhibition of Ces results in changes in odor-mediated OSN activities²⁰ Development of an OR-based volatile odorant sensor allowed us to functionally interrogate the role of xenobiotic enzymes like Ces1d, and its role in modulating specific OR response patterns. We focused on Ces1d, the most abundant Ces expressed in the olfactory mucosa and an ortholog of the human Ces3^{21, 22}, as a model enzyme.

In addition, we have incorporated the reviewer's comment in Discussion (page 16).

There are various components other than xenobiotic metabolizing enzymes in the nasal mucosa. Especially odorant binding proteins, which belong to lipocalin superfamily, are considered to play an important role in odor detection.

Reviewer #2 (Remarks to the Author):

Matsunami's group has been studying interactions of odorant receptors (ORs) and their ligands using a heterologous assay system that they developed. In the present study, the authors performed a large-scale screening to identify ORs that respond robustly to odorants of interest. They also developed a heterologous system capable of discriminating structurally similar odorants in vapor. The authors further characterized an olfactory mucosal enzyme that modulates OR activities. These studies provide us with useful information for our understanding of odor detection and discrimination. Unfortunately, however, this reviewer finds that the paper needs more conceptual advancement to attract the broad audience of Nat. Commun. Thus, I would recommend this paper be published in a more specialized journal.

Response: We hope that the reviewer agrees that our revised manuscript with additional data and analyses will attract the broad audience of Nature Communications.

Specific comments are as follows:

1. In the text, it is not always clear which figure or table is being referred

Response: These errors are corrected.

2. In this study, the authors chose to use seven specific odorants in the first screening. Why were they selected? What was the basis for this selection?

Response: We thank the reviewer for this valuable comment. We have revised the text to expand rationale in Results and Discussion.

In Results, we added the following on page 6 (changes are marked in red).

To identify ORs that robustly respond to odorants *in vitro*, we conducted a large-scale screen of mouse ORs against a panel of seven odorants: acetophenone, cyclohexanone, eugenol, heptanal, 2-heptanone, methyl benzoate and N-amyl acetate (Figure 1A). These odorants, representing diverse functional groups (ester, ketone, allyl benzene and aldehyde) and structures (straight and cyclic aliphatic, and aromatic), are broadly used in the field^{1, 2, 3 4 5, 6, 7, 8, 9, 10}. Further, acetophenone, eugenol and heptanal have well-established cognate ORs, Olfr160,

also known as M72 for acetophenone ¹¹, Olf73, also known as mOR-EG for eugenol ¹² and Olf2, also known as I7 for heptanal ^{13, 14}.

In Discussion, we added a sentence that strengthens rationale on page 13

First we conducted a large-scale screening to identify ORs responding robustly to odorants of interest, including methyl benzoate, the active odor of cocaine ¹⁵ and cyclohexanone, the odor component released from explosives ^{16, 17}

3. Some main figures, e.g., Fig. 1, do not contain sufficient data and are suitable for supplementary figures.

Response:

Fig.1 is not supplementing Fig.2 or any other figures, and we feel that Fig.1 is essential for the readers to follow how our initial screening was conducted. After discussing with the Editor, we have decided to keep Fig. 1 as a part of main figures.

4. The authors should emphasize what is novel in the present system compared with the previous luciferase assay.

Response: We have revised the text as follows on page 15 (changes are marked in red):

In the olfactory system, odorants reach to the olfactory epithelium as volatiles and are dissolved in the olfactory mucus before activating the ORs. In typical heterologous cell based assay systems, however, odor stimulation is performed by replacing the medium with odor-containing medium where odorants are dissolved. Our new assay system aims to better mimic the olfactory system in the heterologous cells by monitoring real time activation events in the presence of volatile odor in the reading chamber.

5. In lines 155-157, the authors describe the toxicity of odorants at high concentrations on Hana cells. What is the evidence for this?

Response: We thank the reviewer for the valuable comment. We have conducted additional experiments and analysis to evaluate the cell viability and observed a dramatic reduction of cell viability with 100% eugenol vapor suggesting high toxicity. In contrast, the majority of the cells survived with 100% methyl benzoate vapor stimulation, suggesting lower toxicity.

We revised the text to expand the Results on page 9 as follow (changes are marked in red).

At 10^0 (undiluted) concentrations, we observed varying degrees of diminished responses. To test the possibility of high odorant concentrations influencing cell viability and toxicity, we conducted the CellTiter-Glo cell viability assay (see methods for details). Eugenol and methyl benzoate were chosen as representative odorants based on our observations that 10^0 eugenol yielded a dramatic reduction in activity whereas 10^0 methyl benzoate yielded only a moderate reduction in activity. A 75% reduction of cell viability was observed with 10^0 eugenol vapor stimulation compared with no odor control after two hours, but a much more moderate reduction of cell viability was observed with 10^0 methyl benzoate (Supplementary Fig. 18). Altogether, these data are consistent with the idea that high odorant concentration influence cell viability and odorant induced responses..

Supplementary Fig. 18
Cell toxicity of odorants exposed to the cells using CellTiter-Glo® Luminescent Cell Viability Assay (Promega)..

The luminescence in each well was measured at 120 min after stimulation by vapor phase odorant. The luminescence was normalized such that the value of each well on before stimulation was defined 1. Error bars indicate s.e.m (n=4). The p-value was calculated with one-way ANOVA followed by Turkey multiple comparison test (**p<0.01, ****p<0.0001).

We also added its procedure in the Material and Method section on page 19 as follows

Cell viability assays

To assess cell viability, Hana3A cells were plated at a 100% confluence in 96-well plates overnight. Cell viability was tested at t=0 and after 2h incubation in the luminometer chamber at room temperature in three different conditions; no odor, 100% methyl benzoate and 100% eugenol (odorant in vapor phase). The ATP content, assessing the cell viability, was monitored using CellTiter-Glo assay (Promega). Before odor treatment, the culturing media was replaced by 25 μ l of HBSS containing 10mM of HEPES and 1mM of D-Glucose. After odor treatment, 25 μ l of CellTiter-Glo Reagent were added to each well and the plate was shaken for 2min and

stabilized at room temperature for 10min. Cell viability was assessed by reading luminescence. Results after 2h incubation were normalized to the t=0 value. Multiple comparisons were performed using one-way analysis of variance (ANOVA) followed by Turkey multiple comparison test.

6. In lines 174-176, the authors emphasize the reproducibility of the present assay. The reproducibility should be compared between the present assay system and the previous one.

Response: We did not intend to demonstrate higher reproducibility of the present assay over previous ones. We apologize for causing confusion. We clarified this point by revising the text as follows on page 10.

Further, we evaluated the reproducibility and discrimination potential of this assay

7. As for the Ces 1d enzyme, how does it work on odor ligands? Which part of ligand molecule is digested?

Response:

Carboxyl esterase metabolizes carboxyl ester groups into alcohols and carboxylic acid. In this paper, we used 3 carboxyl esterase: eugenol acetate, benzyl acetate and N-amyl acetate. Thus, Ces1d used in this study should catalyze the odorants shown in Supplementary Figure 27. Consistent with the prediction, Olfr979, which does not respond to eugenol acetate but eugenol, responds to eugenol acetate when coexpressed with Ces1d (see Figure 7 and Supplementary Figure 27). We added the figure showing the carboxyl esters and the predicted products as Supplementary Figure 27A on page 12 and revised the text as follows on page 12.

Consistent with the enzymatic action of Ces1d, Olfr979, which does not respond to eugenol acetate but eugenol, responded to eugenol acetate when co-expressed with Ces1d (2.8-fold change, FDR corrected p<0.01) (Fig. 7A and Supplementary Fig. 27B).

8. This reviewer feels that the Ces1d treatment should be performed in the first large-scale screening, rather than in the narrowed down OR set.

Response:

Demonstrating Ces1d-dependent changes of responses with a selected set of ORs forms a platform for a large-scale screening. Given the time and resources necessary, conducting the suggested large-scale screening with ~800 ORs with Ces1d is beyond the scope of the current work (our original screening described in Fig.1 took over half a year with a team effort).

Moreover, though such a large-scale screening with Ces1d is likely to yield many additional ORs, it is unlikely these odorant-OR combinations will result in conceptually new findings beyond what we have already shown with the selected ORs. To incorporate the reviewer's valuable suggestion, we added the following in the revised manuscript on page 16:

Furthermore, it will be interesting to perform a large-scale screening in the presence of Ces1d.

9. In lines 280-283, the authors claim that their new assay system directly detects vapor odorants. However, even in this assay, ORs expressed in Hana cells are detecting odorants dissolved in GloSensor cAMP reagent.

Response: The reviewer is correct in that ORs expressed in Hana3A cells are likely to detect vapor odorants after they dissolved into the assay buffer. This is analogous to what is happening in the nose where vapor odorants dissolve into nasal mucus before activating ORs. We revised the manuscript to clarify this point on page 7.

In these assays, the responses were presumably mediated by dissolution of the tested odorant from the vapor phase into the medium, followed by odorant binding and activation of the ORs (Fig. 3A).

Reviewer #3 (Remarks to the Author):

Elucidating the chemical specificities of odorant receptors (ORs) as a group is critical for understanding the coding logic of olfaction. Such knowledge can also be applied to develop so-called biomimetic sensors that can be used as chemical detectors in a broad set of applications. However both goals require the ability to assay the ligand activation profiles of hundreds if not thousands of ORs in a context that recapitulates these receptors' function in vivo, namely to detect volatile chemicals. Thus, in this study Matsunami and colleagues describe a cell-based assay for odorant OR activation in response to odor ligands presented in the vapor phase. The authors initially screened over 800 mammalian ORs against 7 odors in liquid phase in a cell-based assay that reports on receptor activation of cAMP via a CREB-dependent transcription of a luciferase reporter gene. From this screen they winnowed the ORs down to 138 unique receptors showing statistically significant activity over background for further study. Next, the authors used a real-time assay system based on cAMP activation of an engineered luciferase whose activity is dependent on cAMP binding. The assay was then adapted to screen compounds in vapor phase and characterize the specificities of 32 selected ORs. From these analyses the authors argue that their assay is capable of discriminating both structurally divergent as well as similar odor molecules based on the differential activation of ORs within the set.

Over all, this study represents a technical tour de force and may indeed provide the foundation for building a biosensor for volatile chemicals.

However in its current form the manuscript falls a bit short in its claims. My major concern is with the interpretation of the data in Figure 6 on the differential activation of ORs by structural variants of acetophenone. While the authors argue based on their statistical analysis that the repertoire of 32 ORs can distinguish among these compounds, setting aside menthone (ME) as an outlier, a qualitative assessment of the first 10 or so ORs in panel B might lead one to conclude that the responses are more or less the same. If the authors wish to make the argument that their assay can in principle serve as a biosensor capable of discriminating related compounds, they should put this to the test by e.g. using a subset of data as a training set to build a classifier, and then asking whether the classifier can reliably assign the identities of test odors in independent experiments.

Response: We thank the reviewer for the valuable advice. Following the reviewer's suggestions, using a subset of data as a training set to build a classifier, we asked whether the classifier

could reliably assign the identities of test odors in independent experiments. Specifically we built a random forest classifier trained on three replicates of seven odors against 31 ORs and a vector control. We then used the classifiers to predict the odors presented in an independent dataset with the same format. The random forest classifier was 95.2% accurate, with the only error occurring when the classifier predicted BA for receptor responses to HAC.

```
## Confusion Matrix and Statistics
##
##           Reference
## Prediction AC BA HAC MAC ME MS PP
## AC      2  0  0  0  0  0  0
## BA      0  3  0  0  0  0  0
## HAC     0  0  3  0  0  0  0
## MAC     1  0  0  3  0  0  0
## ME     0  0  0  0  3  0  0
## MS     0  0  0  0  0  3  0
## PP     0  0  0  0  0  0  3
##
## Overall Statistics
##
##           Accuracy : 0.9524
##           95% CI : (0.7618, 0.9988)
## No Information Rate : 0.1429
## P-Value [Acc > NIR] : 2.274e-16
##
##           Kappa : 0.9444
## Mcnemar's Test P-Value : NA

## Statistics by Class:
##
##           Class: AC Class: BA Class: HAC Class: MAC Class: ME
## Sensitivity      0.66667  1.0000  1.0000  1.0000  1.0000
## Specificity      1.00000  1.0000  1.0000  0.9444  1.0000
## Pos Pred Value   1.00000  1.0000  1.0000  0.7500  1.0000
## Neg Pred Value   0.94737  1.0000  1.0000  1.0000  1.0000
## Prevalence      0.14286  0.1429  0.1429  0.1429  0.1429
## Detection Rate   0.09524  0.1429  0.1429  0.1429  0.1429
## Detection Prevalence 0.09524  0.1429  0.1429  0.1905  0.1429
## Balanced Accuracy 0.83333  1.0000  1.0000  0.9722  1.0000
##
##           Class: MS Class: PP
## Sensitivity      1.0000  1.0000
## Specificity      1.0000  1.0000
## Pos Pred Value   1.0000  1.0000
## Neg Pred Value   1.0000  1.0000
## Prevalence      0.1429  0.1429
## Detection Rate   0.1429  0.1429
## Detection Prevalence 0.1429  0.1429
## Balanced Accuracy 1.0000  1.0000
```

Please see supplementary Data 7

Fig. 6F t-SNE plot drawn from a random forest classifier trained on three replicates of seven odors against 31 ORs and a vector control.

We have added the text as follows on page 10 (changes are marked in red):

To address whether the repertoire of 31 ORs can distinguish among these compounds, we built a random forest classifier trained on three replicates of seven odors against 31 ORs and a

vector control. Using the classifier to predict the odors presented in an independent dataset with the same format. The random forest classifier was 95.2% accurate, with the only error occurring when the classifier predicted benzyl aldehyde for receptor responses to 2-hydroxy acetophenone (Fig. 6F and Supplementary Data 7).

Minor points:

The authors choose to use area under the curve (as somewhat unfortunate choice of terms, as AUC is typically used in ROC analyses) for the real-time assays to define the level of receptor activation. What is the rationale for doing so vs. using peak response?

Response:

Following the reviewer's suggestions, we compared AUC and peak response for each condition at 10^{-2} . Correlation between AUC and peak response were very high ($R^2 > 0.970$). We next compared the coefficient of variation of the value of AUC or peak response value in each OR/odorant combinations. Coefficient of variation of AUC tend to be lower than that of the peak value, suggesting that OR responses are more stable when AUC is used for analysis. This is likely because we use all the measurements when calculating AUC where as we use single measurements when using peak response.

This analysis is shown in Supplementary Fig. 19 and Supplementary Data 11)

Supplementary Fig. 19

We have added the following sentences in the Materials and Methods section on page 18

We obtained very similar results in OR responses AUC values and peak responses ($R^2 > 0.970$ at 10^{-2}) (Supplementary Fig. S19 and Supplementary Data 11). Well-to-well variations tended to be less when AUC values were used for analysis.

The dynamic range of the real-time luciferase assay seems to be quite low, precluding the ability to generate any meaningful dose-response curve that should be expected to commence and saturate over 2 logs as expected for a pseudo-first-order reaction. While at face value the assay can discriminate compounds, the authors should comment on this limitation.

Response:

ORs that show responses to odorants only at higher concentrations, it is indeed the case that the dynamic range is low because of toxicity of odorants at the highest concentration. We have added the following sentence in Discussion on page 15.

One limitation of the new assay system is that the dynamic range is low for low-affinity ORs due to toxicity of the tested odorant at high concentrations.

In Figure 3, the authors wish to demonstrate that OR activation is comparable whether the compounds are delivered in liquid or vapor phase. While the left-hand plot in panel E does convincingly show a linear relationship, the right-hand plot most decidedly does not; rather it is biphasic or at least show a significant offset during the initial phase of the response. The authors should comment and perhaps temper their claims accordingly.

Response:

The reviewer is correct that the left-hand plot in panel E convincingly shows a linear relationship between 10^{-2} vapor vs 50uM liquid stimulation, the right-hand plot between 10^{-4} vapor vs 50uM liquid stimulation does not, primarily because most of ORs do not respond to acetophenone at 10^{-4} .

We have revised the text on page 8:

Comparing the response of Olfr145 by acetophenone in liquid versus vapor phase stimulation using the GloSensor approach, revealed a significant linear relationship at 50 μ M liquid phase stimulation to 10⁻² vapor phase stimulation ($R^2=0.89$) (Fig. 3E left). Comparison of 50 μ M liquid phase stimulation to 10⁻⁴ vapor phase stimulation also revealed a significant but less-linear relationship ($R^2=0.69$), likely due to no/little response for most ORs at 10⁻⁴ acetophenone. Overall, we observed significant positive correlations for all the tested odorants with varying R^2 (10⁻² dilution: $p<10^{-6}$. $R^2=0.52-0.89$) (10⁻⁴ dilution: $p<2\times 10^{-4}$. $R^2=0.36-0.92$) (Fig. 3E, 3F) (Supplementary Fig. 3 and 4). Altogether, these results demonstrate that our vapor stimulation assay is capable of monitoring OR activity by volatile odorants in real time.

References

1. Bhandawat V, Reisert J, Yau KW. Elementary response of olfactory receptor neurons to odorants. *Science* **308**, 1931-1934 (2005).
2. Goldsmith BR, *et al.* Biomimetic chemical sensors using nanoelectronic readout of olfactory receptor proteins. *ACS Nano* **5**, 5408-5416 (2011).
3. Neuhaus EM, Gisselmann G, Zhang W, Dooley R, Stortkuhl K, Hatt H. Odorant receptor heterodimerization in the olfactory system of *Drosophila melanogaster*. *Nature neuroscience* **8**, 15-17 (2005).
4. Belluscio L, Lodovichi C, Feinstein P, Mombaerts P, Katz LC. Odorant receptors instruct functional circuitry in the mouse olfactory bulb. *Nature* **419**, 296-300 (2002).
5. Cassenaer S, Laurent G. Conditional modulation of spike-timing-dependent plasticity for olfactory learning. *Nature* **482**, 47-52 (2012).
6. Banerjee A, *et al.* An Interglomerular Circuit Gates Glomerular Output and Implements Gain Control in the Mouse Olfactory Bulb. *Neuron* **87**, 193-207 (2015).
7. Xu F, *et al.* Simultaneous activation of mouse main and accessory olfactory bulbs by odors or pheromones. *The Journal of comparative neurology* **489**, 491-500 (2005).
8. Mozell MM, Jagodowicz M. Chromatographic separation of odorants by the nose: retention times measured across in vivo olfactory mucosa. *Science* **181**, 1247-1249 (1973).
9. Pace U, Hanski E, Salomon Y, Lancet D. Odorant-sensitive adenylate cyclase may mediate olfactory reception. *Nature* **316**, 255-258 (1985).
10. Saito H, Chi Q, Zhuang H, Matsunami H, Mainland JD. Odor coding by a mammalian receptor repertoire. *Science signaling* **2**, ra9 (2009).
11. Bozza T, Feinstein P, Zheng C, Mombaerts P. Odorant receptor expression defines functional units in the mouse olfactory system. *The Journal of neuroscience : the official journal of the Society for Neuroscience* **22**, 3033-3043 (2002).

12. Kajiya K, Inaki K, Tanaka M, Haga T, Kataoka H, Touhara K. Molecular bases of odor discrimination: Reconstitution of olfactory receptors that recognize overlapping sets of odorants. *The Journal of neuroscience : the official journal of the Society for Neuroscience* **21**, 6018-6025 (2001).
13. Zhao H, Ivic L, Otaki JM, Hashimoto M, Mikoshiba K, Firestein S. Functional expression of a mammalian odorant receptor. *Science* **279**, 237-242 (1998).
14. Krautwurst D, Yau KW, Reed RR. Identification of ligands for olfactory receptors by functional expression of a receptor library. *Cell* **95**, 917-926 (1998).
15. Furton KG, Caraballo NI, Cerreta MM, Holness HK. Advances in the use of odour as forensic evidence through optimizing and standardizing instruments and canines. *Philos Trans R Soc Lond B Biol Sci* **370**, (2015).
16. Furton KG, Myers LJ. The scientific foundation and efficacy of the use of canines as chemical detectors for explosives. *Talanta* **54**, 487-500 (2001).
17. Ong TH, Mendum T, Geurtsen G, Kelley J, Ostrinskaya A, Kunz R. Use of Mass Spectrometric Vapor Analysis To Improve Canine Explosive Detection Efficiency. *Anal Chem* **89**, 6482-6490 (2017).
18. Olson MJ, Martin JL, LaRosa AC, Brady AN, Pohl LR. Immunohistochemical localization of carboxylesterase in the nasal mucosa of rats. *J Histochem Cytochem* **41**, 307-311 (1993).
19. Bogdanffy MS, Randall HW, Morgan KT. Biochemical quantitation and histochemical localization of carboxylesterase in the nasal passages of the Fischer-344 rat and B6C3F1 mouse. *Toxicol Appl Pharmacol* **88**, 183-194 (1987).
20. Nagashima A, Touhara K. Enzymatic conversion of odorants in nasal mucus affects olfactory glomerular activation patterns and odor perception. *J Neurosci* **30**, 16391-16398 (2010).
21. Olender T, *et al.* The human olfactory transcriptome. *BMC genomics* **17**, 619 (2016).

22. Ibarra-Soria X, Levitin MO, Saraiva LR, Logan DW. The olfactory transcriptomes of mice. *PLoS Genet* **10**, e1004593 (2014).

Responses to Reviewers:

Reviewer #2

Question: One thing that is not clear to me is the ligand specificity and selectivity of ORs in the present screening system compared with those in the previous one using the luciferase assay in Hana cells. Are they basically the same or different for some ligand-OR pairs?

Response:

Odor-mediated responses with our luciferase reporter gene assays in Hana3A cells and Glosensor-based assays described in the current manuscript are similar in a way that strongly activated ORs in one assay responds to the same odorant in the other assay. In order to make it easier to compare between the Luciferase assay screening data and Glosensor assay data, the notation was unified to "Olf" nomenclatures in all data. And we added the following in the revised manuscript on page 9.

Strongly activated ORs in our luciferase reporter gene assays (Supplementary Data 1) responded to the same odorant in the Glosensor assays.

Reviewer #3

Question: The authors have in principle addressed my main concern by building a random forest classifier model to verify that their assay system can make accurate predictions. However the method itself and resulting t-SNE plot in Figure 6F are both inadequately described. These issues should be addressed prior to publication.

Response:

We thank the reviewer for the valuable comment. We have revised the text on page 10 and 11.

To visualize the 32-dimensional representation of the seven odors we used the t-Distributed Stochastic Neighbor Embedding (t-SNE) dimensionality reduction technique (Fig. 6C). (on page 10)

We then tested the same odors against the same ORs in triplicate on a different day and used the classifier to predict the odors presented in this independent dataset. (on page 11)

We also add detailed methods describing how we built classifier in the Methods section on page 20.

Discriminating odors computationally

31 ORs and a vector control were tested against seven odorants. To visualize this 32-dimensional representation of the seven odorants we used the dimensionality reduction technique t-Distributed Stochastic Neighbor Embedding (t-SNE) with the perplexity parameter set to 10. To formally test the ability of the assay data to predict the odorant, we used a random forest classifier. In a random forest, multiple decision trees are built from a random sampling of data with replacement (bootstrap samples). Furthermore, a random set of features are used to determine the best split at each node during the construction of a tree. Similar results were obtained using a linear discriminant analysis, with a single classification error (MAC was predicted for an AC trial). All models were implemented in the R statistical package version 3.5.0.